# Illuminating biological pathways for drug targeting in head and neck squamous cell carcinoma

Gabrielle Choonoo[1,2☯¤a], Aurora S. Blucher[1,3☯*], Samuel Higgins[2¤b], Mitzi Boardman[2], Sophia Jeng[1,4], Christina Zheng[1,2], James Jacobs[1,2,5], Ashley Anderson[6], Steven Chamberlin[2], Nathaniel Evans[2], Myles Vigoda[3,6], Benjamin Cordier[2], Jeffrey W. Tyner[1,3,7], Molly Kulesz-Martin[3,6], Shannon K. McWeeney[1,2,4], Ted Laderas[1,2]

1 Knight Cancer Institute, Oregon Health & Science University, Portland, Oregon, United States of America, 2 Division of Bioinformatics and Computational Biology, Department of Medical Informatics & Clinical Epidemiology, Oregon Health & Science University, Portland, Oregon, United States of America, 3 Department of Cell, Developmental & Cancer Biology, Oregon Health & Science University, Portland, Oregon, United States of America, 4 Oregon Clinical and Translational Research Institute, Oregon Health & Science University, Portland, Oregon, United States of America, 5 Pediatric Hematology and Oncology, OHSU Doernbecher Children's Hospital, Portland, Oregon, United States of America, 6 Department of Dermatology, Oregon Health & Science University, Portland, Oregon, United States of America, 7 Division of Hematology and Medical Oncology, Oregon Health & Science University, Portland, Oregon, United States of America

☯ These authors contributed equally to this work.
¤a Current address: Regeneron Pharmaceuticals, Tarrytown, New York, United States of America
¤b Current address: Roche Sequencing Solutions, Santa Clara, California, United States of America
* blucher@ohsu.edu

## Abstract

Head and neck squamous cell carcinoma (HNSCC) remains a morbid disease with poor prognosis and treatment that typically leaves patients with permanent damage to critical functions such as eating and talking. Currently only three targeted therapies are FDA approved for use in HNSCC, two of which are recently approved immunotherapies. In this work, we identify biological pathways involved with this disease that could potentially be targeted by current FDA approved cancer drugs and thereby expand the pool of potential therapies for use in HNSCC treatment. We analyzed 508 HNSCC patients with sequencing information from the Genomic Data Commons (GDC) database and assessed which biological pathways were significantly enriched for somatic mutations or copy number alterations. We then further classified pathways as either "light" or "dark" to the current reach of FDA-approved cancer drugs using the Cancer Targetome, a compendium of drug-target information. Light pathways are statistically enriched with somatic mutations (or copy number alterations) and contain one or more targets of current FDA-approved cancer drugs, while dark pathways are enriched with somatic mutations (or copy number alterations) but not currently targeted by FDA-approved cancer drugs. Our analyses indicated that approximately 35–38% of disease-specific pathways are in scope for repurposing of current cancer drugs. We further assess light and dark pathways for subgroups of patient tumor samples according to HPV status. The framework of light and dark pathways for HNSCC-enriched biological pathways allows us to better prioritize targeted therapies for further research in HNSCC based

**Data Availability Statement:** The full workflow, including scripts for analyses and figure creation can be found on GitHub:https://github.com/biodev/HNSCC_Notebook. Data are reported within the

paper, its Supporting Information, and the GitHub repository.

**Funding:** This work was supported by the National Cancer Institute 1R01CA192405 (MKM, SKM), National Library of Medicine Informatics Training Grant T15LM007088 (ASB, MB, SC, BC), Knight Cancer Institute P30 CA069533 and the National Center for Advancing Translational Sciences 5UL1TR000128 (SKM). Samuel Higgins is now employed by Roche Sequencing Solutions and Gabrielle Choonoo is now employed by Regeneron Pharmaceuticals. Their contributions to this paper as outlined in the author contributions section were made while they were employed by Oregon Health & Science University, and as such their current employers did not play a role in the study design, data collection and analysis, decision to publish, or preparation of this manuscript.

**Competing interests:** Samuel Higgins is now employed by Roche Sequencing Solutions and Gabrielle Choonoo is now employed by Regeneron Pharmaceuticals. Their affiliation with these companies does not alter our adherence to PLOS ONE policies on sharing data and materials.

on the HNSCC genetic landscape and FDA-approved cancer drug information. We also highlight the importance in the identification of sub-pathways where targeting and cross targeting of other pathways may be most beneficial to predict positive or negative synergy with potential clinical significance. This framework is ideal for precision drug panel development, as well as identification of highly aberrant, untargeted candidates for future drug development.

## Introduction

There are over a half million cases of head and neck cancer diagnosed every year worldwide and almost 90% of these are squamous cell carcinomas (HNSCC). The 5-year survival rate remains around 50% for patients [1]. Most HNSCC cases are not diagnosed until advanced stages of disease, however 30% of cases are caught early and have a higher survival rate [2]. Risk factors for this disease include smoking, alcohol consumption, and human papilloma virus (HPV) [3]. Because of the heterogeneity in tumor types of this disease, the prognosis and survival outcomes vary greatly and in turn require diverse therapy options for patients. For instance, HPV-positive patients diagnosed with HNSCC are a distinct clinically recognized subgroup with a better prognosis than HPV-negative patients [4,5].

Standard treatment for early HNSCC includes surgery or radiation with the addition of chemotherapy for later stages of disease [2,6]. HNSCC is notable among cancers because the tumors occur in structurally complex areas such as the larynx, pharynx, oral cavity, paranasal sinuses and the salivary glands. Therefore, surgery can severely impact the quality of life for the patient due to impairment of critical functions such as speech and swallowing [7,8]. Targeted therapies for cancer treatment can offer more effective and less functionally morbid options, often resulting in overall improved quality of life compared to radiation and chemotherapy treatment. Currently there are three FDA approved targeted therapies for HNSCC including cetuximab which is a humanized antibody to EGFR [9], and pembrolizumab and nivolumab, both of which are PD-1 inhibitors [10]. Of these, cetuximab is the only tumor intrinsic targeted therapy; it is used with radiation therapy for local or advanced tumors and also for recurrent or metastatic tumors [9]. Pembrolizumab and nivolumab are used for patients whose disease worsens during or after platinum chemotherapy [11]. Ongoing clinical trials are investigating the efficacy of these drugs in combination with other targeted and immunotherapies in treating HNSCC [12]. Overall though, HNSCC remains a high-need area with respect to targeted therapies. To address this need, we leverage both the genetic landscape of the disease and current FDA approved drug-target information in the context of biological pathways to assess HNSCC drug-targetable space. Our specific purpose with this work is to identify potentially targetable vulnerabilities in HNSCC cancers that can be followed up with additional preclinical testing for future advancement into translational pipelines and ultimately therapeutic options.

Pathway-based approaches offer several advantages for understanding genetic alterations in disease cohorts [13]. Mapping genetic alterations across a patient cohort to shared dysregulated pathways allows us to focus on key shared characteristics in the cohort which may not be obvious at the gene level [13]. Additionally, mapping gene level alterations to pathways provides greater biological context for how these alterations may be affecting tumor behavior [14]. Pathway approaches often rely on "over-representation analysis" which assesses whether the number of alterations observed for a particular pathway is more than what would be expected

at random [13]. One well known over-representation method is Gene Set Enrichment Analysis, which identifies sets or groups of functionally related genes for which genetic alterations are enriched [15]. Such over-representation methods allow us to identify those pathways likely to be important for a particular disease or treatment state.

The Genomic Data Commons (GDC) contains National Cancer Institute (NCI)-generated data from the most comprehensive cancer genomic datasets including The Cancer Genome Atlas (TCGA), Therapeutically Applicable Research to Generate Effective Therapies (TARGET), and the Cancer Genome Characterization Initiative (CGCI). These datasets contain genomic and clinical data across 33 cancer types and subtypes with the goal of providing the cancer research community a tool for discovering better care for cancer patients at the molecular level [16]. The genomic landscape of HNSCC has previously been found to be characterized by TP53 mutations, whole genome duplications, chromosomal gains and losses affecting cell cycle checkpoints and PI3K-AKT signaling, and increased rates of somatic copy number alterations [8,17–19]. We use the HNSCC clinical, somatic mutation, and copy number data to identify biological pathways involved with diverse tumor types of this disease (Fig 1). We then assess whether these HNSCC biological pathways contain targets of FDA-approved cancer drugs using the Cancer Targetome. The Cancer Targetome is a drug-target resource with supporting evidence levels for all current FDA-approved cancer drugs. We introduce the concept of "light" and "dark" HNSCC-enriched pathways, where "light" pathways are within scope of FDA-approved cancer drugs while "dark" pathways are out of scope of FDA-approved cancer drugs. Light pathways represent targeting opportunities for HNSCC with respect to existing cancer drugs and therefore may offer repurposing opportunities. Dark pathways represent future targeting opportunities for HNSCC as they are currently out of scope of current cancer drugs but found to be of biological relevance to HNSCC.

## Results

### HNSCC patient demographics

For this analysis, we used the most updated version of the GDC clinical HNSCC data, which contains a total of 528 patients. Of these, 508 patients have accompanying somatic mutation data and 296 have accompanying copy number data. We annotated 57 (11%) patients as HPV positive and 118 (23%) as HPV negative. Within this dataset, 333 (66%) patients did not have conclusive HPV results. The mean patient age was 61 years, 73% of patients were male, 85% were white, and 75% of patients smoked or used tobacco.

Most primary tumor tissues (63%) were in the oral cavity, 22% from the larynx, and 15% from the oropharynx. Over half (54%) of the tumors were in stage IV, and 43% of the tumors were in stages I-III, while 3% did not have tumor stage annotation. At the time of study, 57% were living and 43% were deceased with a median follow-up of 541 days. The majority of patients (66%) had radiation treatment with a mean radiation dose of 3,945 centigray (cGy) and most of these were in combination with other therapy types: 29% of patients had both radiation and surgery, 26% had surgery only, 25% had chemotherapy, radiation, and surgery, 6% had chemotherapy and radiation, 3% had radiation only, and 7% did not have therapy annotation. Additionally, 19 patients were categorized as 'other' and had a combination of the standard therapeutics along with immunotherapy or targeted molecular therapy denoting a small percentage of patients that were treated with precision medicine methods.

### HNSCC somatic mutations

For the 507 patients with somatic mutation data, we identified a total of 15,085 unique gene symbols in the mutation data. The number of mutations per patient ranged from 1 to 1,999

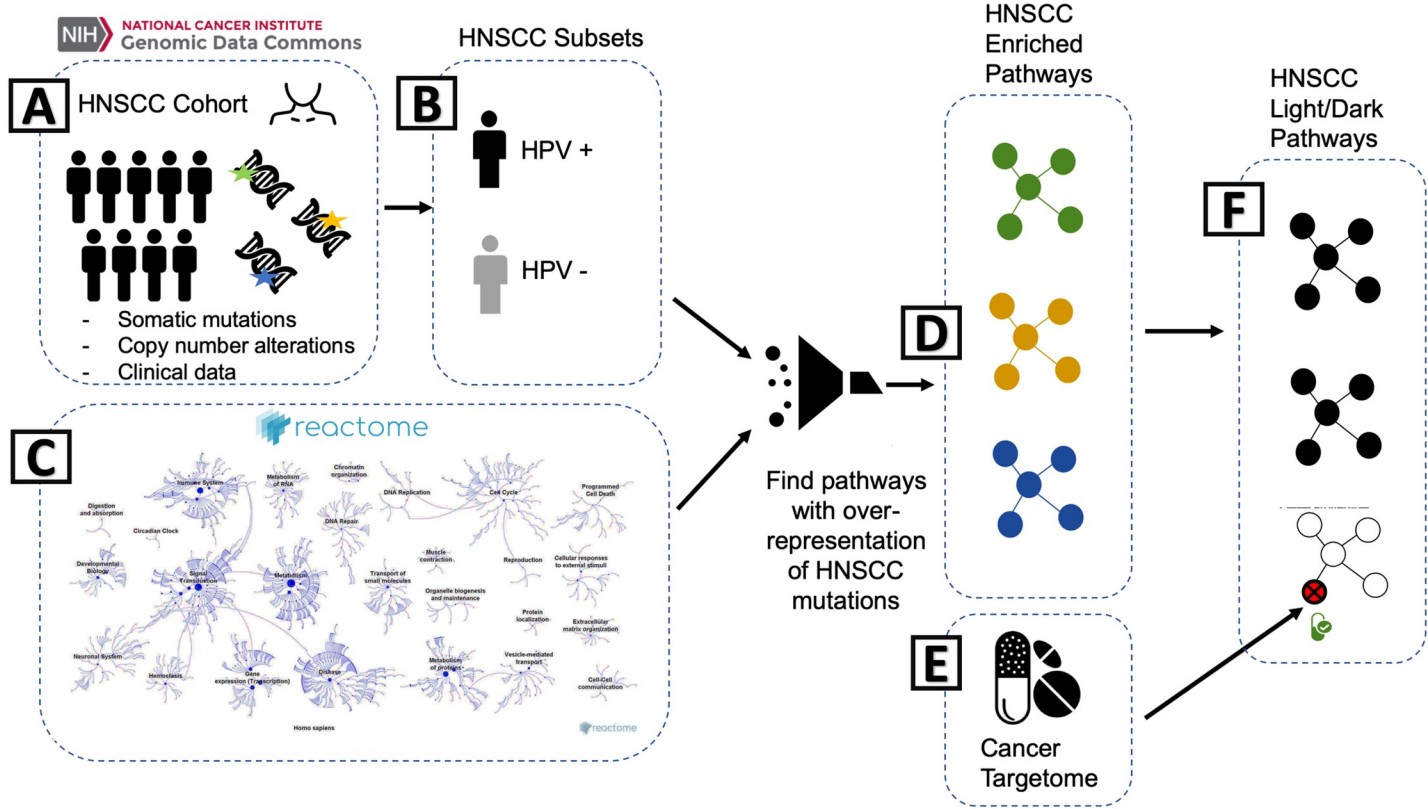

**Fig 1. Identifying Targetable Pathways in the GDC Head and Neck Squamous Cell Carcinoma Cohort. A)** Somatic mutation, copy number alteration, and clinical data for patients with head and neck squamous cell cancer (HNSCC) are selected from the Genomic Data Commons. **B)** HPV status was annotated as described in the methods for a subset of the cohort. **C)** The genes with somatic mutations (or copy number alterations) for HNSCC (represented by green, gold or blue in (A)) are then mapped onto the pathways in the Reactome database. **D)** An overrepresentation analysis is done, using the hypergeometric probability distribution, to identify Reactome pathways likely to be aberrant for HNSCC patients (represented by green, gold and blue pathways). **E)** Proteins and associated FDA-cancer drugs, from the Cancer Targetome database, are then mapped onto the aberrant HNSCC pathways. **F)** A pathway that contains a cancer drug-associated protein is then considered 'light', as designated by the bottom pathway in the box on the right. Pathways with no association to the Cancer Targetome are considered 'dark'.

with a median of 82 and a mean of 117.35. There were a total of 67,302 mutations and most of them were missense mutations (84%). According to the GDC annotation, 85% of these mutations were moderate and about 15% were high impact. The top 20 mutated genes, ranked by the number of mutations per gene symbol, include TP53, TTN, FAT1, MUC16, CSMD3, SYNE1, CDKN2A, LRP1B, NOTCH1, KMT2D, PIK3CA, PCLO, DNAH5, USH2A, FLG, NSD1, RYR2, AHNAK, CASP8, and PKHD1L1 (Fig 2A).

## HNSCC copy number alterations

For the 296 patients with copy number alteration data, we identified a total of 21, 872 unique altered genes. The number of copy number alterations per patient ranged from 1 to 2,812 with a median of 308.5 and a mean of 444. The top 20 copy number altered genes, ranked by number of patients harboring a copy number alteration in the gene, include: HAS_MIR-548K, MIR528K, PPFIA1, ANO1, FADD, CTTN, CDKN2A, FGF3, FGF4, FGF19, CCND1, ORAOV1, CDKN2B, C9ORF53, MYEOV, SHANK2, CDKN2B-AS1, MIR2664, ZMAT3, and ABCC5 (Fig 2B).

## HNSCC light and dark pathways

By mapping HNSCC-specific mutational and copy number aberrations to biological pathways, we aimed to better understand the pathway context and identify druggable space. We queried

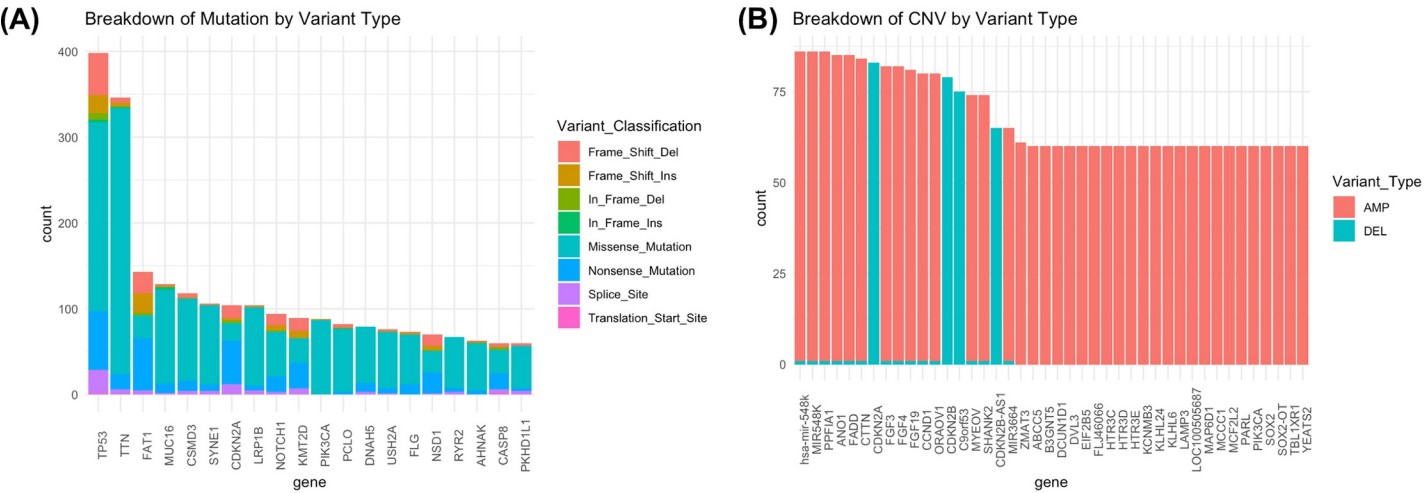

**Fig 2.** A. Top 20 mutated genes in the GDC HNSCC patient cohort (N = 507). We only included variants that had high or moderate impact, which were classified as Missense, Nonsense, Nonstop, Frame Shift Deletion, Frame Shift Insertion, In Frame Deletion, In Frame Insertion, Splice Site and Translation Start Site. We excluded variants with mostly low or modifying impact, which were classified as 3'Flank, 3'UTR, 5'Flank, 5'UTR, IGR, Intron, RNA, Silent, and Splice Regions. B. Top 20 copy number altered genes in the GDC HNSCC patient cohort (N = 296). We only included copy number alterations characterized by "-2" or "+2", for high confidence deletions and amplifications, respectively.

the Reactome biological pathways [20] and calculated (1) over-representation of HNSCC somatic mutations per pathway and (2) over-representation of HNSCC copy number alterations per pathway. For both analyses we used a total of 1650 pathway models covering 7631 genes. Overall for the full cohort of 507 HNSCC patients, we found a total of 15,085 unique mutated genes, 6,195 (41%) of which mapped to pathways and 8,890 (59%) of which did not. From a pathway perspective, a total of 1,638 (99%) pathways contained one or more mutated

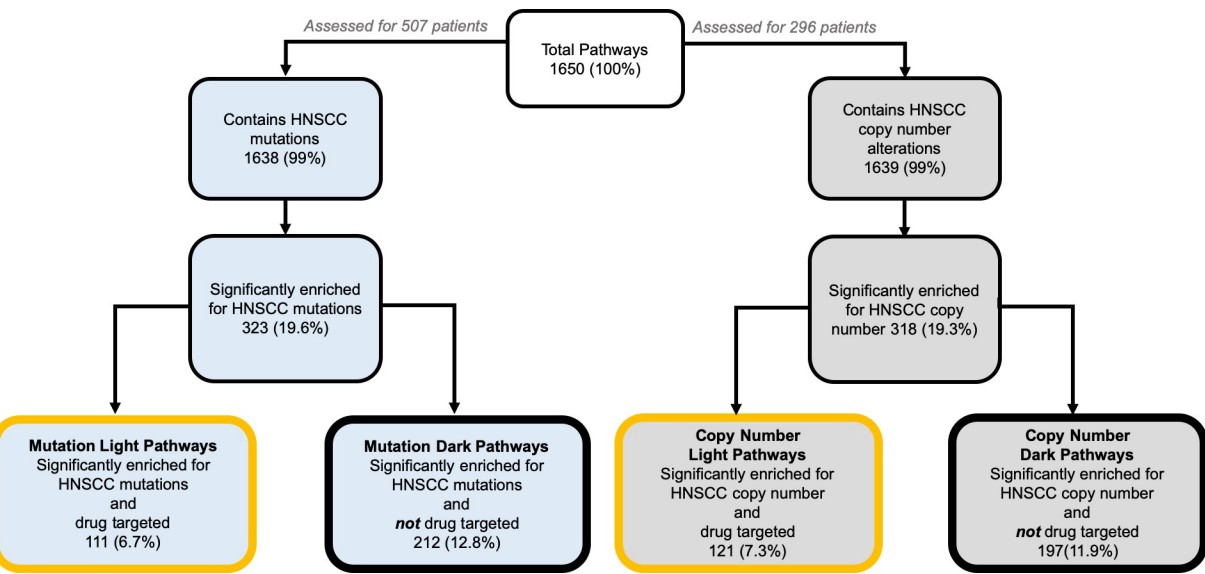

**Fig 3. Light and dark pathway coverage for HNSCC mutation and copy number enriched pathways.** Diagram of the subsets of Reactome pathways in HNSCC that are mutation-enriched or copy-number alteration enriched, with drug-targeted subsets labeled for each. Percentages shown in the figure are out of the total number pathways used for this analysis (1650). Of the HNSCC-specific mutation enriched pathways, 34% are targetable by FDA approved cancer drugs (light pathways) and 66% are open for drug development (dark pathways). Of the HNSCC-specific enriched pathways, 38% are targetable by FDA-approved cancer drugs (light pathways) and 62% are open for drug development (dark pathways.

genes. Of these, we found a total of 323 (19.6%) pathways to be significantly enriched for mutated genes in HNSCC (Fig 3). For the copy number analyses, which covered 296 patients in the HNSCC cohort, we found a total of 21,872 altered genes, 6,649 (30%) which mapped to pathways and 15,223 (70%) that did not. By assessing pathways according to an over-representation of HNSCC alterations, we can find higher level shared dysregulation among patients in the HNSCC cohort that may not necessarily be apparent when considering gene-level alterations alone.

We then assessed whether these pathways contained drug targets according to Cancer Targetome information [21]. We designated pathways containing one or more drug targets as "light" pathways, i.e. within scope of current FDA-approved cancer drugs. Pathways containing no drug targets were designated as "dark" or outside the scope of current cancer drugs. Out of the 323 pathways significantly enriched for HNSCC mutations, 111 were light to cancer drugs (S1 Table), while 197 were dark (S2 Table). Out of the total number of biological pathways used in this analysis, approximately 7% were both mutation enriched in the HNSCC cohort and light to current cancer drugs, while approximately 13% were mutation enriched in the HNSCC cohort but dark to current cancer drugs (Fig 3). Out of the 318 pathways significantly enriched for HNSCC copy number alterations, 121 were light to cancer drugs (S3 Table), while 197 were dark (S4 Table). Out of the total number of pathways used in analysis, approximately 7% were copy number enriched and light to cancer drugs, while 12% were copy number enriched but dark (Fig 3). Interestingly, using either data type results in a comparable number of HNSCC-specific enriched pathways.

If we consider light and dark pathways as a fraction of the total number of HNSCC mutation enriched pathways, approximately 34% are light while 66% are dark. For the HNSCC copy number enriched pathways, approximately 38% are light while 62% are dark. We found that a comparable percentage of mutation and copy number enriched pathways are potentially light to current FDA-approved drugs. On the one hand, for a disease like HNSCC which needs more therapy options, the finding that approximately 35–38% of disease-specific pathways may be potentially in scope of cancer drugs presents an enormous targeting opportunity. The HNSCC mutation enriched light pathways range in size from 1 to 618 genes, with 86–100% of their gene members mutated throughout the cohort. The percentage of gene members targeted within mutation enriched light pathways ranges from <1% to 100%, with between 1 and 87 targets of FDA-approved cancer drugs. The HNSCC copy number enriched light pathways range in size from 1 to 1164, with 90–100% of their gene members have copy number alterations throughout the cohort. The percentage of gene members targeted within copy number enriched light pathways ranges from 3% to 100%, with between 1 and 59 targets of FDA-approved cancer drugs.

On the other hand, the 62–65% of HNSCC-enriched pathways that are dark to cancer drugs present an opportunity for developing new chemical compounds or repurposing from non-cancer domains. These HNSCC mutation enriched dark pathways range in size from 1 to 111 genes and range in coverage of mutated genes from 93–100%. Thus dark pathways are comparable to light pathways with regards to extent of enrichment for HNSCC patient mutations. The HNSCC copy number enriched dark pathways range in size from 1 to 29 and all have a 100% coverage of copy-number altered genes across the cohort.

Assessment of the overlap of enriched pathways between the two data types revealed that 43 pathways are enriched in both mutations and copy number alterations as well as light to FDA-approved cancer drugs (Fig 4). Similar assessment revealed that 112 pathways are enriched in both mutation and copy number alterations and are dark to FDA-approved cancer drugs. Thus, these pathways are of particular interest for drug repurposing or future drug development, respectively, as they are supported by multiple data types in the cohort.

## Prioritization of HNSCC light and dark pathways

By further prioritizing both the light and dark pathways, we can better focus efforts on drug targeting for HNSCC that takes into account the diverse genetic landscape of this disease. Prioritization of light and dark pathways for further analysis can consider many factors- here we focus on prioritization based on mutational load on pathways and the extent to which the pathway is affected in the overall cohort. Pathway pathogenicity, or percent of pathway mutated, provides a measure of the extent to which a pathway is affected across many samples. Out of our 111 mutation enriched light pathways, 82 had 100% of their gene members mutated across the HNSCC cohort. We then ranked according to proportion of the HNSCC cohort with one or more gene mutations in the pathway. In Table 1, we show the top 15 light ranked pathways according to these two metrics.

As an example of a top-ranked light HNSCC-enriched pathway, we highlight 'Nephrin Interactions'(https://reactome.org/content/detail/R-HSA-373753) (Fig 5). For this pathway, all 22 genes are covered by HNSCC mutations from different samples in the cohort, and the cohort-centric perspective shows that 36% of all patients have one or more gene mutations in the pathway. With respect to drug-targeting, this pathway has a target coverage of 18%. A total of 6 drugs were found to interact with targets in this pathway, with supporting binding assay evidence from the Cancer Targetome of <1000nM: bosutinib, crizotinib, dasatinib, idelalisib, sunitinib, and vandetanib (Table 2).

Nephrin (NPHS1) is mostly expressed in kidney podocytes and is an important component of the adherens junction between these kidney cells [22]. However, the relevance of this pathway to HNSCC is likely due to the PI3K-AKT signaling cascade, which is directly downstream of (and initiated by) NPHS1 tyrosine phosphorylation from the Src kinase FYN.

Previous studies have found the PI3K pathway to be the most highly mutated oncogenic signaling pathway in HNSCC: 12.6% of HNSCC tumors were found to have a mutation in the PI3KCA gene itself, with 30.5% of tumors carrying a mutation in the pathway and 6.6% carrying multiple mutations in the PI3KCA pathway [23]. Furthermore, targeting the PI3K pathway with PI3K inhibitors in mutation enriched tumors was found to be effective in vitro and in vivo [23]. This exemplifies how pathway analysis leads to rational drug selection in light pathways [23].

Similarly, when we rank mutation enriched dark pathways according to pathway pathogenicity, 186 had 100% of their gene members mutated across the HNSCC cohort. We also rank dark pathways according to proportion of HNSCC cohort with one or more gene mutations in the pathway (Table 3). We note that the top-ranked dark pathways are found to be mutated in the HNSCC cohort to a similar extent as the light pathways, highlighting their attractiveness for future therapy development. The top ranked dark mutation enriched pathway is "Collagen biosynthesis and modifying enzymes", which has 64 pathway members altered throughout the cohort, and is altered in approximately 52% of the patients in the cohort. NOTCH-related signaling pathways make up 4 of the top 15 dark pathways; NOTCH1 is well-known for harboring inactivating mutations in HNSCC [24,25]. Targeting such loss-of-function mutations is an area of both continued difficulty and great promise in drug development.

We can also prioritize copy number enriched light and dark pathways in a similar manner. The top 15 copy number light pathways are shown in Table 4 and the top 15 copy number dark pathways are shown in Table 5. Out of our 121 light pathways, 115 had 100% of their gene members exhibit a copy number alteration across the HNSCC cohort. For the top light copy number enriched pathways, we see that a higher proportion of the cohort is represented by the altered pathways compared to the top light mutation enriched pathways. This likely reflects that HNSCC is known to be a copy number alteration heavy disease [26]. Iron uptake

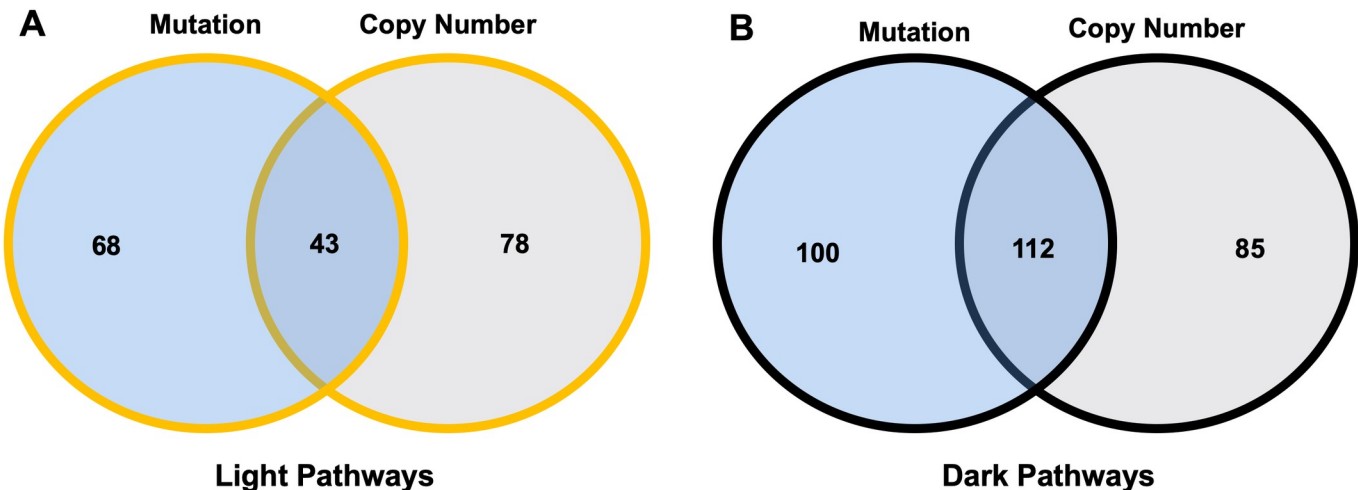

**Fig 4. Overlap of light and dark pathway coverage between data types.** Left Venn diagram shows overlap of mutation-enriched and copy-number enriched light pathways. Right Venn diagram shows overlap of mutation-enriched and copy number enriched dark pathways.

and transport is the top ranked copy number enriched light pathway (Table 4), which is of interest given recent discussion on the potential for targeting alterations in iron homeostasis in HNSCC [27,28]. We also note that 'Nephrin Interactions', highlighted in Fig 5 as a mutation enriched light pathway, is also in our top 15 ranked copy number enriched light pathways.

Out of our 197 dark pathways, all 197 had 100% of their gene members exhibit a copy number in 1 or more patients. For dark copy number enriched pathways, we also see a similar

**Table 1. Top 15 light HNSCC mutation enriched pathways.**

| Pathway | Number of Mutated Genes | Proportion Pathway Mutated | Proportion of HNSCC cohort with ≥1 mutated gene in pathway | Number of patients with ≥1 mutated gene in pathway |
|---|---|---|---|---|
| Nephrin interactions | 22 | 1 | 0.35502959 | 180 |
| Constitutive Signaling by Ligand-Responsive EGFR Cancer Variants | 19 | 1 | 0.28796844 | 146 |
| Constitutive Signaling by EGFRvIII | 15 | 1 | 0.2800789 | 142 |
| TRP channels | 25 | 1 | 0.24457594 | 124 |
| Ras activation uopn Ca2+ infux through NMDA receptor | 17 | 1 | 0.23076923 | 117 |
| Unblocking of NMDA receptor, glutamate binding and activation | 17 | 1 | 0.21893491 | 111 |
| Kinesins | 27 | 1 | 0.21696252 | 110 |
| Caspase-mediated cleavage of cytoskeletal proteins | 12 | 1 | 0.20512821 | 104 |
| Sema3A PAK dependent Axon repulsion | 16 | 1 | 0.19329389 | 98 |
| CRMPs in Sema3A signaling | 16 | 1 | 0.16962525 | 86 |
| DSCAM interactions | 11 | 1 | 0.15384615 | 78 |
| Growth hormone receptor signaling | 24 | 1 | 0.14201183 | 72 |
| Mitotic Telophase/Cytokinesis | 14 | 1 | 0.13806706 | 70 |
| Na+/Cl- dependent neurotransmitter transporters | 19 | 1 | 0.12031558 | 61 |
| Recycling of bile acids and salts | 15 | 1 | 0.11637081 | 59 |

Light pathways ranked first according to proportion of pathway mutated and second according to proportion of cohort with mutated gene.

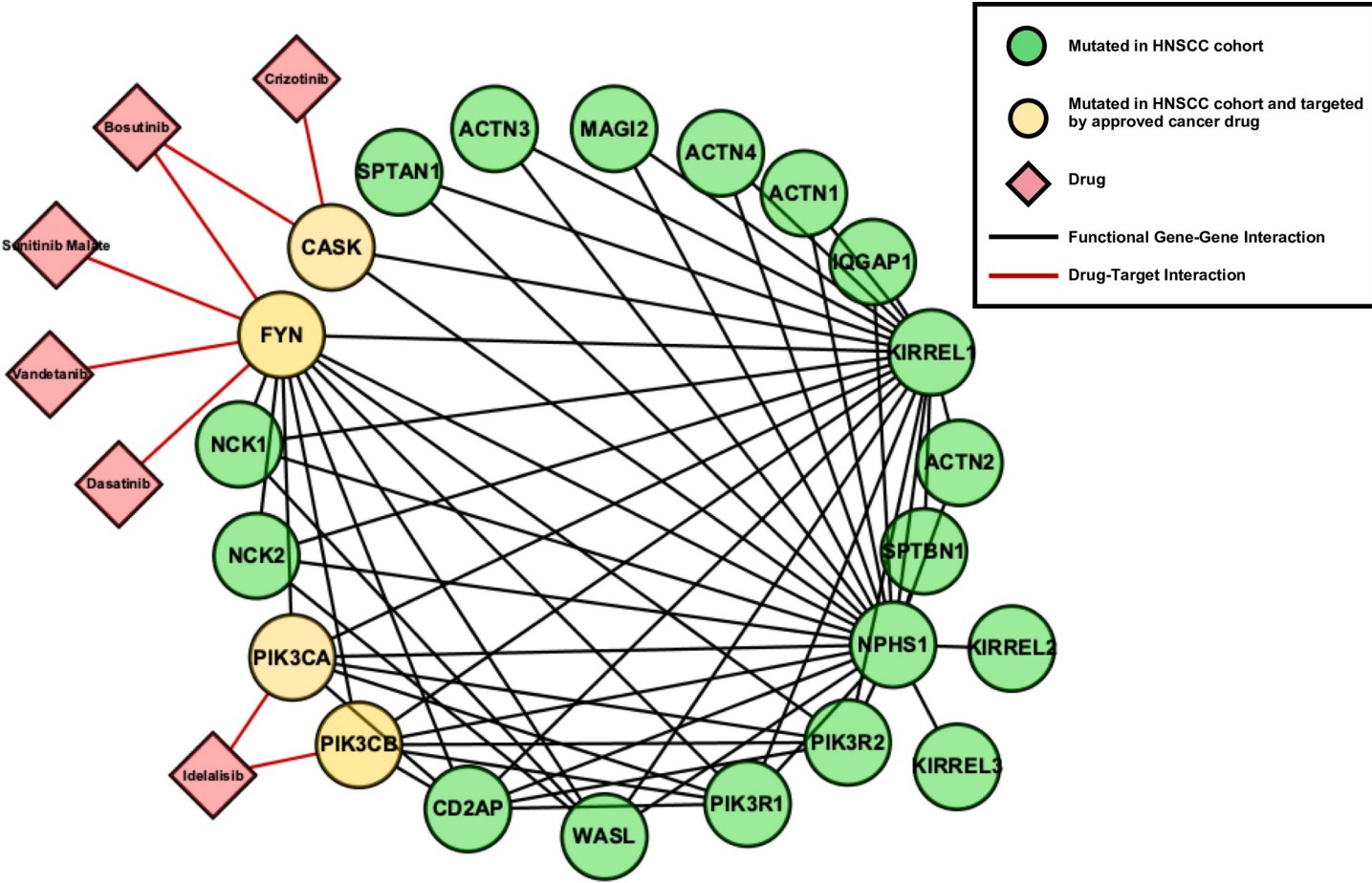

**Fig 5. The pathway "Nephrin Interactions" is highly aberrant in the GDC HNSCC cohort and Light to FDA-approved Cancer Drugs.** Nephrin Interactions is an example of a top-ranked light pathway, defined as a pathway highly covered with HNSCC mutations (~100%) and frequently mutated in the patient cohort (36%). Nodes in green are genes mutated in the GDC HNSCC cohort and nodes in yellow are mutated in the cohort as well annotated as targets for cancer drugs in the Cancer Targetome. Drugs are indicated by red diamonds and represent FDA-approved cancer drugs with targets in this pathway. For drug-target interactions shown here, we required supporting binding assay evidence to be <1000nM.

proportion of the cohort represented as found for the dark mutation enriched pathways. The TRAIL signaling pathway, which is the top ranked copy number enriched dark pathway (Table 5), has previously been found to be an apoptotic regulator in HNSCC [29,30]

**Table 2. Targets in the Nephrin Interactions Pathway Hit by Drugs in the Cancer Targetome.**

| Target | Drug | Binding Assay Type | Binding Assay Value (nM) |
|--------|------|--------------------|--------------------------|
| CASK | Bosutinib | KD | 830 |
| CASK | Crizotinib | KD | 140 |
| FYN | Bosutinib | KD | 11 |
| FYN | Bosutinib | IC50 | 1.799999952 |
| FYN | Dasatinib | KD | 0.79 |
| FYN | Sunitinib Malate | KD | 520 |
| FYN | Vandetanib | KD | 360 |
| PIK3CA | Idelalisib | IC50 | 820 |
| PIK3CB | Idelalisib | IC50 | 562 |

For each drug-target interaction, the best (minimum) binding assay value is shown in nM units. Assay types may be dissociation constant (KD) or IC50.

**Table 3. Top 15 dark HNSCC mutation-enriched pathways.**

| Pathway | Number of Mutated Genes | Proportion Pathway Mutated | Proportion of HNSCC cohort with ≥1 mutated gene in pathway | Number of patients with ≥1 mutated gene in pathway |
|---|---|---|---|---|
| Collagen biosynthesis and modifying enzymes | 64 | 1 | 0.52071006 | 264 |
| Laminin interactions | 23 | 1 | 0.32741617 | 166 |
| NICD traffics to nucleus | 13 | 1 | 0.25641026 | 130 |
| Notch-HLH transcription pathway | 13 | 1 | 0.25641026 | 130 |
| Receptor-ligand binding initiates the second proteolytic cleavage of Notch receptor | 14 | 1 | 0.23865878 | 121 |
| Platelet calcium homeostasis | 19 | 1 | 0.21301775 | 108 |
| Loss of Function of FBXW7 in Cancer and NOTCH1 Signaling | 5 | 1 | 0.17751479 | 90 |
| Adenylate cyclase activating pathway | 10 | 1 | 0.17554241 | 89 |
| Constitutive Signaling by NOTCH1 t(7;9) (NOTCH1:M1580_K2555) Translocation Mutant | 7 | 1 | 0.17357002 | 88 |
| Vitamin D (calciferol) metabolism | 7 | 1 | 0.16370809 | 83 |
| Dermatan sulfate biosynthesis | 11 | 1 | 0.13609467 | 69 |
| GABA A receptor activation | 13 | 1 | 0.13412229 | 68 |
| regulation of FZD by ubiquitination | 21 | 1 | 0.13412229 | 68 |
| Reduction of cytosolic Ca++ levels | 10 | 1 | 0.12820513 | 65 |
| CHL1 interactions | 9 | 1 | 0.12426036 | 63 |

Dark pathways ranked first according to proportion of pathway mutated and second according to proportion of cohort with mutated gene.

## HPV cohort

A substantial subset of HNSCC cancers are associated with human papillomaviruses (HPVs) and have been found to be biologically and clinically distinct from non-HPV associated

**Table 4. Top 15 light HNSCC Copy number enriched pathways.**

| Pathway | Number of Genes with Copy Number Alteration | Proportion Pathway with Alteration | Proportion of HNSCC cohort with ≥1 altered gene in pathway | Number of patients with ≥1 altered gene in pathway |
|---|---|---|---|---|
| Iron uptake and transport | 43 | 1 | 0.47635135 | 141 |
| Dimerization of procaspase-8 | 11 | 1 | 0.4527027 | 134 |
| Regulation by c-FLIP | 11 | 1 | 0.4527027 | 134 |
| Cholesterol biosynthesis | 22 | 1 | 0.43581081 | 129 |
| NF-kB activation through FADD/RIP-1 pathway mediated by caspase-8 and -10 | 12 | 1 | 0.41216216 | 122 |
| PLC-gamma1 signalling | 33 | 1 | 0.40202703 | 119 |
| Signaling by FGFR1 fusion mutants | 19 | 1 | 0.39527027 | 117 |
| VEGFR2 mediated vascular permeability | 26 | 1 | 0.39189189 | 116 |
| TRIF-mediated programmed cell death | 10 | 1 | 0.38851351 | 115 |
| DAG and IP3 signaling | 31 | 1 | 0.38851351 | 115 |
| Ca-dependent events | 28 | 1 | 0.38513514 | 114 |
| Nephrin interactions | 22 | 1 | 0.375 | 111 |
| Gap junction trafficking and regulation | 30 | 1 | 0.37162162 | 110 |
| CaM pathway | 26 | 1 | 0.37162162 | 110 |
| Calmodulin induced events | 26 | 1 | 0.37162162 | 110 |

Light pathways ranked first according to proportion of pathway members with copy number alteration and second according to proportion of cohort with copy number altered gene members.

**Table 5. Top 15 dark HNSCC Copy number enriched pathways.**

| Pathway | Number of Genes with Copy Number Alteration | Proportion Pathway with Alteration | Proportion of HNSCC cohort with ≥1 altered gene in pathway | Number of patients with ≥1 altered gene in pathway |
|---|---|---|---|---|
| TRAIL signaling | 7 | 1 | 0.43918919 | 130 |
| Transferrin endocytosis and recycling | 29 | 1 | 0.43918919 | 130 |
| Gap junction trafficking | 28 | 1 | 0.36486486 | 108 |
| Insulin-like Growth Factor-2 mRNA Binding Proteins (IGF2BPs/IMPs/VICKZs) bind RNA | 8 | 1 | 0.34121622 | 101 |
| Formation of annular gap junctions | 9 | 1 | 0.3277027 | 97 |
| Gap junction degradation | 10 | 1 | 0.3277027 | 97 |
| FasL/ CD95L signaling | 5 | 1 | 0.31756757 | 94 |
| Recycling of eIF2:GDP | 8 | 1 | 0.2972973 | 88 |
| Fanconi Anemia pathway | 24 | 1 | 0.2972973 | 88 |
| Biotin transport and metabolism | 11 | 1 | 0.29391892 | 87 |
| Hyaluronan biosynthesis and export | 4 | 1 | 0.28040541 | 83 |
| Utilization of Ketone Bodies | 3 | 1 | 0.25337838 | 75 |
| Adenylate cyclase inhibitory pathway | 14 | 1 | 0.25337838 | 75 |
| Inhibition of adenylate cyclase pathway | 14 | 1 | 0.25337838 | 75 |
| Adenylate cyclase activating pathway | 10 | 1 | 0.23310811 | 69 |

Dark pathways ranked first according to proportion of pathway members with copy number alteration and second according to proportion of cohort with copy number altered gene members.

HNSCC cancers [31,32]. Given this information, it is of interest to stratify our pathway analysis with respect to HPV status [31]. For this sub-analyses, we have assessed only with respect to mutation data. The HNSCC cohort with consensus, high-confidence HPV status annotation included 57 HPV positive patients and 118 HPV negative patients. For the HPV positive cohort, we identified 25 light pathways and 6 dark pathways (S5 and S6 Tables). For the HPV negative cohort, we identified 28 light pathways and 27 dark pathways (S7 and S8 Tables). When comparing the pathway analyses between HPV positive and HPV negative groups, 4 pathways were identified as dark to both groups of patients: ABH3 mediated Reversal of Alkylation Damage, Biosynthesis of A2E implicated in retinal degradation, Laminin interactions, and Collagen biosynthesis and modifying enzymes. Conversely, 11 pathways with shared targets were identified as light to both groups.

For the HPV positive cohort, the proportion of light pathways that are found to be mutated ranges from 30% to 100%, and the proportion of the cohort affects for these light pathways ranges from 1% to 87.7%. For dark pathways, the proportion of pathways found to be mutated ranges from 42% to 100%, and the proportion of the cohort affected for these dark pathways ranges from 1% to 31.5%.

For the HPV negative cohort, the proportion of light pathways that are found to be mutated ranges from 44% to 100%, and the proportion of the cohort affected ranges from <1% to 99%. For the dark pathways, the proportion of pathways found to be mutated ranges from 68% to 100%, and the proportion of the cohort affected ranges from <1% to 50%.

## Nested pathway architecture

The pathways in the Reactome database for homo sapiens have a hierarchical structure where the highest level refers to major biological processes such as Cell Cycle, Developmental Biology, Metabolism, Neuronal System, Signal Transduction, etc. Each of these pathways contain "nested" subpathways within it. Here, we denote parent pathways as the top of the hierarchical

structure and child pathways as any subsequent pathways that are contained within the parent pathway.

It is important that we consider this nested architecture in our analysis, as it is possible that dark pathways can actually be nested within higher level light pathways. An example of this is the "GABA A receptor activation" pathway, which is a dark pathway nested within 3 pathways. The parent pathway is "GABA receptor activation", which itself was not a significantly enriched pathway. This pathway is however nested within another parent pathway, "Neuro-transmitter Receptor Binding and Downstream Events", which is a light pathway. This is further nested within the "Transmission across Chemical Synapses", which is a light pathway contained in "Neuronal System".

## Discussion

Further development in targeted therapy options for HNSCC treatment is a high-need area given that current standard of care often leaves patients with loss of critical functions. Here we present our computational approach to explore the potential drug-targetable space in HNSCC based on the known HNSCC genetic landscape. This work fits into the larger context of precision medicine by aiding translational efforts such as the design of cancer-specific drug panels and nominating targets and pathways for further bench testing. While previous approaches have investigated HNSCC from the perspective of multiple data types [6,19] and/or pathway-level aberrations [23], our approach offers a unique perspective in uniting mutation and copy number data with drug-target interactions to investigate shared pathway dysregulation in HNSCC with respect to strength of evidence for drug targeting capabilities.

We have identified biological pathways relevant to this disease using both somatic mutation and copy number alteration data and stratified pathways according to whether or not they are currently in scope of FDA-approved drugs. Light pathways offer a strategy for targeting HNSCC-relevant pathways given the set of currently FDA-approved drugs and thus could potentially bring treatment options to patients through an accelerated route. Our workflow uniquely leverages the Cancer Targetome for drug-target interactions, which allows us to consider all possible targets with which a given FDA-approved drug may interact, according to supporting evidence. This differs from other analyses of targetable pathways in cancer, which often focus on the presumed, or so-called primary targets of drugs or on the class of targets with which a drug interacts [17,33]. Additionally, because the Cancer Targetome provides experimental binding assay values for each drug-target interaction, we can prioritize light pathways according to the strength of the interaction between a drug and its target in the pathway. This transparent and accessible link to drug-target interaction evidence is uniquely enabled by the Cancer Targetome drug-target interaction evidence framework [21].

Additionally, our workflow identifies dark pathways, which also offer valuable information for targeting HNSCC pathways, as they indicate pathways likely to have disease importance that are currently not in scope of cancer drugs. Thus, dark pathways present opportunities for further HNSCC research and drug development that will be of interest to the precision therapy community. It is of note that the Fanconi Anemia pathway was among the top 15 copy number-enriched dark pathways (found in nearly 30% of the cohort) given that head and neck cancer risk is high in patients with Fanconi Anemia a rare condition caused by inherited FANC gene family mutations [34–36].

Additionally, we have also stratified this analysis by HPV tumor status and highlighted how these subgroups have differentially affected light and dark pathways. While previous work has established that HPV-positive and HPV-negative subsets of HNSCC cancers are biologically distinct [32,37], and shown different therapeutic vulnerabilities and/or resistance mechanisms

[38], our work here explicates the differences between these two subsets within a pathway context. This analysis demonstrates that different HNSCC tumor types may benefit from specific targeted therapies and further, suggests strategic avenues for selecting combination therapies. Indeed, other approaches, such as the network approach taken by Eckhardt, et al. to investigate the HPV-host protein-protein interaction network are now starting to explore vulnerabilities unique to HPV status subsets for future targeting [39].

With regards to prioritization of targets, our approach offers both light and dark pathways as a means to broaden potential to target tumor vulnerabilities [40], or to devise strategies for synthetic lethality within connected pathways [41,42] in order to spur progress toward rational combination therapies. One of the major obstacles of precision medicine is that tumors evolve to resist drug therapy by single agent drug targeted therapies resulting in recurrence of disease. One future direction of this work could be to pursue combinations of drugs that provide dual targeting of both light and dark pathways in HNSCC. Targeting different genes of interest in multiple pathways, at once or sequentially, may allow us to circumvent drug resistance. Alternatively, such approaches may allow us to identify additional routes of druggability after resistance has occurred. Kurtz et al. described a discovery platform for functional assessment of the effectiveness of combination therapies [43].

We hypothesize that molecular therapeutics for HNSCC can be expanded by a rational approach combining *in silico* evaluation of GDC genomics data and functional analysis of HNSCC cell response to inhibitor panels. Our approach considers mutational load, copy number alteration load of pathways (significantly enriched pathways) and potential druggability of pathways (light pathways to cancer drugs). Future directions for this work could include the application to expression data to find pathways enriched for over or under-expressed genes as potentially of interest for targeting. We note that by considering drug-target interactions for all FDA-approved cancer drugs, we potentially open up the space to consider repurposing drugs from other cancer areas to treat HNSCC. This work could also be expanded to consider non-cancer drugs for potential repurposing into the cancer domain to treat HNSCC. For instance, recent work by Hedberg, et al. found that the use of nonsteroidal anti-inflammatory drugs was associated with improved outcomes in PIK3CA-mutated head and neck cancer [44]. Furthermore, a particularly attractive future direction for pathway analyses of this nature would be to identify pathway vulnerabilities for potential targeting with both targeted therapies and immunotherapies, which are an area of active interest and ongoing clinical trials [31]. While this work has focused on HNSCC, our approach is highly relevant for translational efforts moving forward with cancer-specific therapies, in particular the design of cancer-specific drug screening panels. We anticipate that it is especially relevant for other heterogeneous cancers, in which case pathway approaches can potentially find shared dysregulation that may not be apparent at gene-only level.

In our commitment to open source and reproducible workflows, we have leveraged only open-source resources in this computational framework, including the GDC data portal, Reactome pathways, and the Cancer Targetome. Additionally, we have made our full workflow available through GitHub to encourage dissemination in the precision medicine community.

## Methods

### Study design and data selection

The light and dark pathway study design leverages public -omics data from the GDC and public drug target interaction information to understand greater biological pathway context. For the GDC data, we identified the most significantly mutationally aberrant pathways in HNSCC. These HNSCC-related pathways were then examined for known drug targets in order to guide

drug panel development to evaluate potential drug repurposing for HNSCC treatment. In addition to those pathways currently targeted by existing drugs, defined as "light" pathways, we detected and quantified the number of "dark" pathways that are not currently targeted to guide future drug development.

We analyzed open access Mutect2 HNSCC somatic mutations from the GDC Data Portal that used Illumina Hiseq 2000 and Miseq to sequence primary and metastatic HNSCC tumor samples and mapped to the human genome version 38 (time stamped March 16[th] 2017). We processed the Gene Symbols within the somatic mutation data by checking for approved or synonym nomenclature. We then queried the cleaned, unique Gene Symbols mutated for each patient cohort within the Reactome gene-pathway membership data for homo sapiens.

The GDC annotated the impact of mutations based on Ensembl classification of severity of the variant consequence [20]. The categories are high, moderate, low, and modifier impact variants. High impact means the variant is assumed to have disruptive impact in the protein, most likely causing protein truncation, loss of function or triggering nonsense mediated decay. Moderate impact variants are non-disruptive variants that might change protein effectiveness. Low impact is assumed to be mostly harmless and unlikely to change protein behavior. Lastly, modifier variants are usually non-coding or variants affecting non-coding genes, where there is no evidence of impact. For our analysis, we included variants that were moderate and high impact variants.

For copy number, we analyzed open access GISTIC HNSCC copy number aberrations from the GDC Data Portal. We used high confidence copy number alteration calls of -2 and +2, which correspond to high confidence deletions and amplifications, respectively.

For the patient cohort stratification, we used the HNSCC clinical data from the GDC Data Portal (time stamped May 9[th], 2017) for the 508 patients with somatic mutation data. We collapsed the original anatomic neoplasm subdivision column into three categories: Oral Cavity, Oropharynx, and Larynx. The Oral Cavity category includes: Alveolar Ridge, Buccal Mucosa, Floor of mouth, Hard Palate, Hypopharynx, Lip, Oral Cavity, and Oral Tongue. The Oropharynx category includes Base of tongue and Tonsil. Larynx was its own group.

## HPV tumor status annotation

HPV tumor status annotation was collected from 3 sources: TCGA provisional HPV annotation in the clinical data from cBioPortal [45], the TCGA HNSCC HPV annotation data from the TCGA 2015 Nature publication [26], and the TCGA HNSCC HPV annotation by Nulton et al. 2017 [46]. For our analysis, we classified tumor status as HPV positive or negative according to primarily the Nulton et al. annotation, supported by the other two sources. Nulton et. al. analyzed the raw RNA-Seq and Whole Genome Sequence data (WGS) from TCGA and annotated patient tumors evaluating the expression of viral genes including oncogenes E6 and E7 [46].

Where Nulton et al. annotated tumors as HPV positive and whole genome sequencing was available, we classified these tumors as "Highest Confidence Positive". Additionally, if tumors were annotated as HPV positive by Nulton et al. and also annotated as HPV positive in the TCGA clinical data or in the TCGA Nature publication, we also classified these tumors as "Highest Confidence Positive", even if they did not have whole genome sequencing available. In the case where Nulton et al. annotated tumors as HPV positive that were not annotated in the TCGA sets as positive and no whole genome sequencing was available, we classified these tumors as "High Confidence Positive". Where Nulton et al. annotated tumors as HPV negative and whole genome sequencing was available, we classified these tumors as "High Confidence Negative". For our analysis, we used only "Highest Confidence Positive" cases as HPV positives (n = 57) and "High Confidence Negative" cases as HPV negatives (n = 118).

## Light and dark pathways

For the pathway enrichment analysis and QA/QC, we used in-house workflows in the R Statistical Programming environment using the hypergeometric statistical analysis [47]. This statistical test takes into account the total number of pathways to be evaluated, the total number of genes in those pathways, the number of genes belonging to a pathway, and of those genes, the number of genes that are aberrational in the patient cohort. The p-value is calculated by taking 1 minus this formula. Pathways were considered aberrationally enriched based on the FDR adjusted p-value < 0.05. We used the Benjamini & Yekutieli (BY) method of adjustment [48]. The light and dark pathways were categorized based on whether or not these aberrationally enriched pathways contained gene members that are drug targeted genes from the Cancer Targetome. The top pathways were ranked based on the percentage of the pathway covered with HNSCC aberrations, the percentage of the cohort that contained at least one mutated gene in the pathway, and the number of aberrational genes in the pathway. This takes into account pathway pathogenicity, pathway size and frequency in patients.

## Cancer Targetome

For drug-target interaction information, we used the Cancer Targetome, curated by Blucher et. al. which aggregates information across four databases including DrugBank, the Therapeutic Targets Database, the IUPHAR/BPS Guide to Pharmacology, and BindingDB and assigns evidence levels reflecting the strength of supporting evidence [21]. The Cancer Targetome has a total of 658 targets and 137 drugs. We used http://www.uniprot.org/uploadlists/ to convert Uniprot IDs to Gene Names. Out of the 658 total targets, 654 mapped to gene symbols. Each drug to target interaction was annotated with an evidence level of I, II, or III. Level I interactions have no additional literature information, Level II has supporting literature information, and Level III has supporting literature information with at least one experimental binding value. Since each unique drug to target interactions can have multiple levels of evidence, we used the maximum (or strongest) level of supporting evidence for each target when examining the levels of evidence for drug targets in each pathway for this study.

## Drug targeted pathways

Out of the 654 targets, 451 (69%) were found in pathways and 197 (31%) did not map. When annotating targets in top pathways, we used the maximum evidence level for each target. Of the 451 that mapped to pathways, 84% had a max evidence level of III, 15% had a max level of II, and only 1% had a max level of I. Supporting literature information and/or experimental binding values denote the evidence level tiers.

Out of the 451 targets that mapped to pathways, there was an overlap of 243 that were found in both significant and non-significant pathways. 202 were unique to non-significant pathways and 6 were unique to significant pathways relevant to HNSCC. There were a total of 120 drugs that target the 451 genes found in pathways. All 120 drugs target non-significant pathways and 93 of them target both non-significant and significant pathways. Of the 93 drugs that target significant pathways, 30 of them were Kinase Inhibitors, 6 were Alkylating Drugs, 31 were inhibitors of Microtubule, Histone Deacetylase, Topoisomerase, Androgen Receptor, Aromatose, Epidermal Growth Factor Receptor, Nucleoside Metabolic, and Vinca Alkaloids.

## Visualization using ReactomeFIViz

Pathway visualization in Fig 4 was created using the Cytoscape app, ReactomeFIViz [14,49,50]. Nephrin interactions was visualized from the perspective of Reactome's Functional Interaction

network, which highlights functional relationships between genes in the pathway. ReactomeFI-Viz integrates drug-target interaction information from the Cancer Targetome and allows for visualization in the context of the functional interaction network. For 'Nephrin Interactions', we selected all drug-target interactions with supporting binding assay evidence <1000nM to include in the visualization.

## Software availability

The full workflow, including scripts for analyses and figure creation can be found on GitHub: https://github.com/biodev/HNSCC_Notebook. The code is generalizable and can easily be modified for other cancers besides HNSCC.

## Supporting information

**S1 Table. HNSCC mutation enriched light pathways.** All significant mutation enriched (FDR<0.05) pathways that contain drug targets.
(XLSX)

**S2 Table. HNSCC mutation enriched dark pathways.** All significant mutation enriched (FDR<0.05) pathways that do not contain drug targets.
(XLSX)

**S3 Table. HNSCC copy number enriched light pathways.** All significant copy number enriched (FDR<0.05) pathways that contain drug targets.
(XLSX)

**S4 Table. HNSCC copy number enriched dark pathways.** All significant mutation enriched (FDR<0.05) pathways that do not contain drug targets.
(XLSX)

**S5 Table. HNSCC HPV-positive mutation enriched light pathways.** For HPV-positive cohort only, all significant mutation enriched (FDR<0.05) pathways that contain drug targets.
(XLSX)

**S6 Table. HNSCC HPV-positive mutation enriched dark pathways.** For HPV-positive cohort only, all significant mutation enriched (FDR<0.05) pathways that do not contain drug targets.
(XLSX)

**S7 Table. HNSCC HPV-negative mutation enriched light pathways.** For HPV-negative cohort only, all significant mutation enriched (FDR<0.05) pathways that contain drug targets.
(XLSX)

**S8 Table. HNSCC HPV-negative mutation enriched dark pathways.** For HPV-negative cohort only, all significant mutation enriched (FDR<0.05) pathways that do not contain drug targets.
(XLSX)

## Author Contributions

**Conceptualization:** Molly Kulesz-Martin, Shannon K. McWeeney, Ted Laderas.

**Data curation:** Gabrielle Choonoo, Mitzi Boardman, Ted Laderas.

**Formal analysis:** Gabrielle Choonoo, Aurora S. Blucher, Myles Vigoda, Ted Laderas.

**Funding acquisition:** Molly Kulesz-Martin, Shannon K. McWeeney.

**Methodology:** Gabrielle Choonoo, Aurora S. Blucher, Mitzi Boardman, Sophia Jeng, Christina Zheng, James Jacobs, Ashley Anderson, Steven Chamberlin, Benjamin Cordier, Jeffrey W. Tyner, Shannon K. McWeeney, Ted Laderas.

**Software:** Gabrielle Choonoo, Aurora S. Blucher, Samuel Higgins, Shannon K. McWeeney, Ted Laderas.

**Supervision:** Molly Kulesz-Martin, Ted Laderas.

**Visualization:** Gabrielle Choonoo, Aurora S. Blucher, Steven Chamberlin, Nathaniel Evans.

**Writing – original draft:** Gabrielle Choonoo, Aurora S. Blucher, Mitzi Boardman, Ashley Anderson, Molly Kulesz-Martin, Shannon K. McWeeney, Ted Laderas.

**Writing – review & editing:** Aurora S. Blucher, Samuel Higgins, Mitzi Boardman, Sophia Jeng, Christina Zheng, James Jacobs, Ashley Anderson, Steven Chamberlin, Nathaniel Evans, Myles Vigoda, Benjamin Cordier, Jeffrey W. Tyner, Molly Kulesz-Martin, Shannon K. McWeeney, Ted Laderas.

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
