## [Decision Letter · Decision Letter 0]

2 Aug 2019

PONE-D-19-19753

Illuminating Biological Pathways for Drug Targeting in Head and Neck Squamous Cell Carcinoma

PLOS ONE

Dear Dr. Blucher,

Thank you for submitting your manuscript to PLOS ONE. After careful consideration, we feel that it has merit but does not fully meet PLOS ONE’s publication criteria as it currently stands. Therefore, we invite you to submit a revised version of the manuscript that addresses the points raised during the review process.

ACADEMIC EDITOR: The authors pull data from available public head and neck squamous cell carcinoma sequencing and cancer drug databases to characterize "light" and "dark" pathways, highlighting opportunities for drug development in "dark" pathways. Overall, there are interesting concepts in the manuscript, but it needs significant revision. I agree with Reviewer 2 that the analysis without copy number variation significantly limits the impact of the work. A significant amount of mutations in HNSCC are CNVs; indeed EGFR, one of the most established aberrant genes in HNSCC and the target of cetuximab (as the authors note), is not pulled out by this analysis as its alterations generally are of CNV and expression changes. Also PIK3CA, as the researchers highlight, has significant CNV aberrations. Similar research by other groups often includes CNV. I would recommend the authors include CNV/expression analysis, or describe their rationale otherwise. Additionally, as Reviewer 2 notes, there has been a fair amount of previous research in this topic. The authors do have potentially novel angles to explore, but should do so in the context of discussions on previous approaches and how the approaches of the authors are unique and add novelty.

The authors should address address major comments 2-4 by reviewer 2. Addressing major comment 1 by reviewer 2 would significantly strengthen the author's research if possible.

We would appreciate receiving your revised manuscript by September 16, 2019. To enhance the reproducibility of your results, we recommend that if applicable you deposit your laboratory protocols in protocols.io, where a protocol can be assigned its own identifier (DOI) such that it can be cited independently in the future. For instructions see: http://journals.plos.org/plosone/s/submission-guidelines#loc-laboratory-protocols

We look forward to receiving your revised manuscript.

Kind regards,

Andrew Birkeland, M.D.

Academic Editor

PLOS ONE

Journal Requirements:

[I have read the journal's policy and the authors of this manuscript have the following competing interests:

GC is now a data analyst for Regeneron Pharmaceuticals, Inc.].   

We note that one or more of the authors are employed by a commercial company: 'Roche Sequencing Solutions' and 'Regeneron Pharmaceuticals, Inc'.

Reviewers' comments:

Reviewer's Responses to Questions

**Comments to the Author**

1. Is the manuscript technically sound, and do the data support the conclusions?

Reviewer #1: Yes

Reviewer #2: Partly

2. Has the statistical analysis been performed appropriately and rigorously? 

Reviewer #1: Yes

Reviewer #2: Yes

3. Have the authors made all data underlying the findings in their manuscript fully available?

Reviewer #1: Yes

Reviewer #2: Yes

4. Is the manuscript presented in an intelligible fashion and written in standard English?

Reviewer #1: Yes

Reviewer #2: Yes

5. Review Comments to the Author

Reviewer #1: Here, the authors present a timely analysis correlating sequencing data of head and neck cancer subtypes mutational loads with its associated "drugome". The authors submit robust analyses, which well articulate their workflow, results, and significance of their work. I anticipate these data are of high value to otolayrngologists, and translational researchers developing drugs for head and neck cancers. Such work warrants a commentary piece in this reviewer's opinion.

Reviewer #2: The authors use a systems biology approach with publicly available data to discern "novel" drug targets in both HPV negative and HPV positive HNSCC based on genetics and drug binding data. Overall, this manuscript has compiled a fairly superficial analysis of public data, which, at first appearance, has already been completed in a similar fashion by other groups, and which is limited to just somatic mutations. In this case, the team applies a unique statistical approach to highlight potential pathways as light or dark, and chose to highlight the PI3K pathway as evidence that the approach was successful (which is already being evaluated in HNSCC clinical trials).

Major comments:

1) Demonstrate that this approach has novelty and can make an impact. The authors need to perform some functional validation, or use data from publicly available drug testing databases, in HNSCC models, to show that this approach adds to existing approaches already used in the literature.

2) Enhance the discussion to compare and contrast this approach to similar previous approaches, reference existing literature on the topic.

3) Why did the authors chose to ignore copy number and/or expression data when designing this approach? Other approaches have successfully implemented additional data for lead target optimization.

4) The writing needs to be improved. The authors only have 20 references and simply gloss over important references and details related to this work, for example, in this section "Targeting the PI3K pathway with PI3K inhibitors in enriched tumors was effective in vitro and in vivo". No references.

Minor comments:

1) Figure quality could be improved substantially, especially the mutation rate figure.

6. PLOS authors have the option to publish the peer review history of their article (what does this mean?). If published, this will include your full peer review and any attached files.

Reviewer #1: No

Reviewer #2: No

---

## [Author Response · Author response to Decision Letter 0]

16 Sep 2019

Please find our response to reviewers uploaded as a separate file, along with our revised manuscript.

---

## [Decision Letter · Decision Letter 1]

26 Sep 2019

Illuminating Biological Pathways for Drug Targeting in Head and Neck Squamous Cell Carcinoma

PONE-D-19-19753R1

Dear Dr. Blucher,

We are pleased to inform you that your manuscript has been judged scientifically suitable for publication and will be formally accepted for publication once it complies with all outstanding technical requirements.

With kind regards,

Andrew Birkeland, M.D.

Academic Editor

PLOS ONE

Additional Editor Comments (optional):

The authors have strengthened their manuscript with the addition of integrated copy number analysis. They do provide a new approach to analysis of existing genomic datasets that can provide value to the head and neck cancer research community. Overall, the manuscript is well-written.

Reviewers' comments:

Reviewer's Responses to Questions

**Comments to the Author**

1. If the authors have adequately addressed your comments raised in a previous round of review and you feel that this manuscript is now acceptable for publication, you may indicate that here to bypass the “Comments to the Author” section, enter your conflict of interest statement in the “Confidential to Editor” section, and submit your "Accept" recommendation.

Reviewer #1: All comments have been addressed

Reviewer #2: All comments have been addressed

2. Is the manuscript technically sound, and do the data support the conclusions?

Reviewer #1: Yes

Reviewer #2: Yes

3. Has the statistical analysis been performed appropriately and rigorously? 

Reviewer #1: Yes

Reviewer #2: Yes

4. Have the authors made all data underlying the findings in their manuscript fully available?

Reviewer #1: Yes

Reviewer #2: Yes

5. Is the manuscript presented in an intelligible fashion and written in standard English?

Reviewer #1: Yes

Reviewer #2: Yes

6. Review Comments to the Author

Reviewer #1: Acceptable for publication in its current form, no further revisions requested. Very interesting work.

Reviewer #2: The authors have addressed my primary concerns (except for substantial functional validation). The manuscript has been improved and adds an additional account of head and neck cancer genetics from publicly available databases to the literature.

7. PLOS authors have the option to publish the peer review history of their article (what does this mean?). If published, this will include your full peer review and any attached files.

Reviewer #1: No

Reviewer #2: No

---

## [Editor Report · Acceptance letter]

1 Oct 2019

PONE-D-19-19753R1 

Illuminating Biological Pathways for Drug Targeting in Head and Neck Squamous Cell Carcinoma 

Dear Dr. Blucher:

I am pleased to inform you that your manuscript has been deemed suitable for publication in PLOS ONE. Congratulations! Your manuscript is now with our production department. 

With kind regards,

on behalf of

Dr. Andrew Birkeland 

Academic Editor

PLOS ONE